# Plant Proteins: Methods of Quality Assessment and the Human Health Benefits of Pulses

**DOI:** 10.3390/foods12152816

**Published:** 2023-07-25

**Authors:** Matthew G. Nosworthy, Gerardo Medina, Zhan-Hui Lu, James D. House

**Affiliations:** 1Guelph Research and Development Center, Agriculture and Agri-Food Canada, Guelph, ON N1G 5C9, Canada; zhanhui.lu@agr.gc.ca; 2College of Pharmacy and Nutrition, University of Saskatchewan, Saskatoon, SK S7N 5E5, Canada; 3Children’s Hospital of Eastern Ontario Research Institute, Ottawa, ON K1H 5B2, Canada; gmedina@cheo.on.ca; 4Department of Food and Human Nutritional Sciences, University of Manitoba, Winnipeg, MB R3T 2N2, Canada; james.house@umanitoba.ca; 5Richardson Centre for Food Technology and Research, 196 Innovation Drive, Winnipeg, MB R3T 2N2, Canada; 6Department of Animal Science, University of Manitoba, Winnipeg, MB R3T 2N2, Canada

**Keywords:** protein quality, PER, PDCAAS, DIAAS, health benefits of pulses, plant-based protein

## Abstract

As countries increase their standard of living and individual income levels rise, there is a concomitant increase in the demand for animal-based protein. However, there are alternative sources. One of the alternatives available is that of increased direct human consumption of plant proteins. The quality of a dietary protein is an important consideration when discussing the merits of one protein source over another. The three most commonly used methods to express protein quality are the protein efficiency ratio (PER), a weight gain measurement; protein digestibility-corrected amino acid score (PDCAAS); and the digestible indispensable amino acid score (DIAAS). The possibility that alterations in the quality and quantity of protein in the diet could generate specific health outcomes is one being actively researched. Plant-based proteins may have additional beneficial properties for human health when compared to animal protein sources, including reductions in risk factors for cardiovascular disease and contributions to increased satiety. In this paper, the methods for the determination of protein quality and the potential beneficial qualities of plant proteins to human health will be described.

## 1. Introduction

The United Nations has projected that by 2050, the global population will surpass 9.7 billion individuals [1]. This rise in the population will increase the strain on multiple industries involved in supplying nutritional foodstuffs, from general production to shipping and distribution, However, attention must also be directed to consumer demand for specific macronutrients. The source of the protein being consumed plays an important role in nutrient availability as animal-based proteins are more digestible and provide sufficient quantities of all the necessary amino acids, while plant-based proteins contain anti-nutritive factors that limit digestibility and are limiting in amino acids such as lysine, tryptophan, cysteine, and methionine [2]. In North America, the quality of a protein source, and the regulatory basis on how content claims are assigned, is based on measurements of growth, i.e., the protein efficiency ratio (PER), or protein digestibility and amino acid content, i.e., the protein digestibility-corrected amino acid score (PDCAAS) [2]. Interestingly, plant-based proteins are being investigated for other properties beyond nutritional quality, as the consumption of plant proteins has been effective in ameliorating pathological conditions such as cardiovascular disease, diabetes, and the dysregulation of lipid metabolism, among others [3,4,5]. In this paper, the methods for the determination of protein quality and the potential beneficial qualities of plant proteins to human health will be described (Figure 1).

### Plant Protein Sources

Although plants are the primary source of protein for the majority of the global population, there are a number of complicating factors preventing the increased utilization of this resource. The overall protein content of plant matter is much lower than that of animals [6]. This forces individuals to either consume greater quantities of plants to achieve a similar protein intake, or extract and concentrate plant-based protein, which can be financially expensive. The complication of less retrievable protein is compounded by the fact that protein derived from plant matter also tends to be limiting in essential amino acids and tends to have a lower digestibility due to the presence of anti-nutritive factors, an issue not found in animal-based sources. Cereals such as oats and the flours of rice, corn, and wheat contain proteins that are limiting in lysine; navy/pinto beans and split yellow peas are limiting in tryptophan; while other plant proteins are limited by their sulfur amino acid content, such as lentils and black/red kidney beans [7]. In order to consume 10 g of equivalent quality protein, it would take 32.2 g of roasted chicken breast (USDA 5063), 79.5 g of boiled eggs (USDA 1129), 63 g of chickpea flour (USDA 16157), 187 g of hulled barley (USDA 20075), and 113 g of an optimized chickpea–barley blend. This would result in a caloric intake of 53 kcal for chicken, 123 kcal for egg, 244 kcal for chickpea, 636 kcal for barley, and 401 kcal for the optimized blend. This indicates that while consumption of an equivalent quantity of quality protein requires ingestion of greater amounts of plant protein, food blending may be an avenue for reducing overall caloric intake as demonstrated by lower intake in the blend compared to hulled barley alone.

There are also a number of health benefits resulting from the consumption of a plant-based diet. Studies have indicated that a primarily plant-based diet is effective for weight loss [8] and weight management [9]. This is due to the low energy density of the plant matter combined with increased satiety after consuming a plant-based meal [10]. Advantages also include a reduced risk of diabetes in the vegetarian population [11], accelerated recovery from coronary disease if a plant-based diet is adopted after diagnosis [12], and a reduction in blood pressure [13]. These benefits are the consequence of a broad ‘plant-based’ diet, but there are a wide variety of plants available for consumption each with different nutritional qualities and prevalence in global production/consumption rates. 

## 2. Determination of Protein Quality and Regulation of Protein Content Claims

Over the past 35 years, there have been many international meetings held to discuss the appropriate methods for the determination of protein quality [7,14,15]. These meetings have produced multiple methods for the calculation of protein quality via the assignment of appropriate reference values and a standardized experimental protocol. The three most commonly used methods today are PER, a weight gain measurement; PDCAAS; and the digestible indispensable amino acid score (DIAAS). Unlike PER, PDCAAS and DIAAS rely upon the determination of both the amino acid composition and digestibility to determine the quality of a protein source. 

### 2.1. Protein Efficiency Ratio

The initial method that was standardized for the determination of protein quality was the protein efficiency ratio (PER). The calculation of PER requires an animal feeding trial to determine protein quality [16]. Diets comprised of 10% crude protein provided exclusively from the test article are fed to weanling rats for a period of four weeks, with diet intake and weight gain being regularly monitored. In addition to any test protein sources, a diet containing 10% casein is also included in the trial as a reference, or control. At the completion of the trial, total weight gain is divided by the total protein consumed to generate the PER. In order to standardize PER across research laboratories, and to reduce inter-run variation, the calculated PER is then adjusted relative to the PER generated by the casein diet as follows: adjusted PER_sample_ = PER_sample_ × (2.5/PER_casein_). The regulation used in determining protein quality labels in Canada is the Protein Rating, which is determined through the multiplication of the adjusted PER by the grams of protein in the reasonable daily intake of the food [17]. If the final protein rating is at least 20, the food qualifies for a ‘source’ claim, and if the rating is at least 40 the food qualifies for a ‘high source’ claim. Although there is no direct relationship between the direct calculation of PER and PDCAAS, Health Canada does allow for the estimation of PER from PDCAAS measurements. This calculation is a rough estimate, PER = PDCAAS × 2.5, and is based on the concept of a complete protein having a PDCAAS score of 1 and PER value of 2.5. Examples of adjusted PER values are provided in Table 1. 

### 2.2. Protein Digestibility Corrected Amino Acid Score

The Joint FAO/WHO Expert Consultation, held in 1989, concluded that the protein digestibility-corrected amino acid score (PDCAAS), expressed as a percent, would be the preferred method of measuring protein quality [7]. Shortly afterwards, this method of determining protein quality was adopted in the United States as the required method for protein quality determination. The calculation of PDCAAS involves multiplying the amino acid score (AAS) by the true fecal nitrogen digestibility (TFD%), as determined in rats. The AAS is determined by comparing the amino acid content of the test protein sources with a reference pattern outlined by the FAO/WHO consisting of the following quantities in mg/g protein: Thr-34, Val-35, Met + Cys-25, Ile-28, Leu-66, Phe + Tyr- 63, His-19, Lys-58, and Trp-11 [7]. If the resulting amino acid ratio is 1.00 or greater, that amino acid is not deficient in the test protein. The lowest amino acid ratio resulting from this comparison is selected as the AAS. The determination of the TFD% requires analyzing the fecal nitrogen content and subtracting that value from the quantity of nitrogen consumed, resulting in an estimation of protein digested and subsequently absorbed. It is important to note that any endogenously produced protein that accumulates in the feces is accounted for by feeding control animals a protein-free diet. However, there is a concern that the use of a protein-free diet also alters the normal metabolism of the animal. The feeding of a protein-free diet, and subsequent fecal nitrogen estimation, generates an endogenous protein correction value, as any protein contained in the feces of those animals must have come from the animal itself and not the diet. Examples of PDCAAS values are provided in Table 1. With respect to protein content claims, the PDCAAS values for a food not intended as a sole source of food for infants must be >0.2, and the values are subsequently multiplied by the reference amount customarily consumed (RACC). If the resulting PDCAAS-corrected protein value is between 5 and 10 g per RACC, the food qualifies as a ‘good source’ of protein, while if the final value is greater than 10 g per RACC, the food qualifies as an ‘excellent source’ of protein [20]. 

### 2.3. Digestible Indispensable Amino Acid Score

The method for determining DIAAS is similar to that of PDCAAS in that it requires the determination of the amino acid composition, and subsequent score, of the test protein source as well as a measure of digestibility [14]. The reference amino acid pattern for DIAAS is as follows, in mg/g protein: Thr-31, Val-43, Met + Cys-27, Ile-32, Leu-66, Phe + Tyr- 52, His-20, Lys-57, and Trp-8.5. This update to the reference pattern increased the requirements for Val by 18.6%, Met + Cys 7.4%, Ile 12.5%, and His by 5%. Conversely, the requirements for Thr were lowered by 9.7%, Phe + Tyr 21.2%, Lys 1.8%, and Trp 29.4%. The calculation itself is as follows: 100 x [(mg of digestible dietary indispensable amino acid in 1 g of the dietary protein)/(mg of the same dietary indispensable amino acid in 1 g of the reference protein)]. Examples of DIAAS values for selected animal and plant proteins are provided in Table 1. However, there are multiple important differences between PDCAAS and DIAAS. In addition to the development of a different reference pattern for DIAAS, the calculation of digestibility was shifted from using fecal nitrogen digestibility to ileal amino acid digestibility. The rationale for using ileal amino acid digestibility has multiple facets including that fecal protein determination contains significant quantities of bacterial contaminants, and that ileal amino acid analysis would consider individual amino acid digestibility rather than a single protein measurement [14]. Content claims under the DIAAS system are similar to that of PDCAAS, where 5–10 g per RACC are ‘good sources’ and greater than 10 g are ‘excellent sources’, However, in order to qualify for a claim, the DIAAS value has to be >75, rather than the 0.2 required for PDCAAS. While DIAAS has yet to be adopted by any jurisdiction for the regulation of protein content claims, the discussion regarding the potential impact on human health and dietary regulations following its adoption has begun [18,20,21]. 

### 2.4. Advantages and Disadvantages of PER, PDCAAS, and DIAAS

The advantages and disadvantages of these protein quality measurements is summarized in Table 2. One of the advantages of the PER, when compared to PDCAAS and DIAAS, is that this value can be determined easily, using only weight gain and protein intake [16]. The other primary advantage is that PER is a growth measurement, providing information regarding the impact of a protein source on weight gain as a function of unit protein consumed, whereas PDCAAS and DIAAS do not provide any information in that regard [7,14,16]. It is important, however, to also consider the disadvantages of the PER method. In order to control for inter-run and inter-laboratory variability, the experimentally derived PER value for a protein source is adjusted according to the experimentally derived PER value for casein, an adjusted value of 2.5, which was used as a control diet [16]. Although this does allow for direct comparison across experimental trials and research laboratories, the adjusted PER provides information relative to the casein diets, not direct values for the protein ingredient itself. This method of determining protein quality also makes the assumption that all consumed protein is directed towards growth, and not maintenance, with an additional complication being that growth is defined as weight gain, without clarification regarding whether that weight is gained through fat accumulation, muscle mass, water, etc. This method is also a rodent-based assay, and, as rats have a higher demand for dietary sulfur amino acids than humans, proteins lower in cysteine and methionine will have a more direct effect on the growth of a rat, and the resulting PER value, than an equivalent diet in a human. 

The PDCAAS measurement of protein quality is beneficial in that it provides detailed information regarding the amino acid composition and fecal digestibility of a protein source [7]. This information, when combined with a reference pattern of human amino acid requirements, can be used to determine whether a protein can provide a full complement of amino acids, or whether the protein source should be combined with another to meet human nutritional requirements. PDCAAS also has the advantage of being mathematically accessible. When considering blending two or more protein sources, the PER must be performed on all mixed proteins, as PER is not additive. PDCAAS data, on the other hand, can be used to generate theoretical blends that show a good relationship with data generated in vivo. Although PDCAAS has been widely adopted and utilized, certain disadvantages have been determined including the truncation of PDCAAS scores and using fecal digestibility in the calculation [22]. Regarding the truncation of PDCAAS scores, it is important to note that this occurs for the comparison of an individual protein source to another; when calculating the PDCAAS of a blend, the amino acid content and protein digestibility data are used and not a truncated PDCAAS value, although the final PDCAAS value will be truncated. Despite the adjustment of PER values according to the casein control, it is possible to have scores greater than 2.5, whereas PDCAAS values are limited to 100, although in some cases the ‘true’ score could be greater. The determination of protein digestibility by using fecal nitrogen can result in an overestimation of PDCAAS values, as there is interaction of the digesta with the colonic microflora which can lead to contamination with microbial nitrogen.

The most recent method for protein quality determination, DIAAS, is a modification of the PDCAAS method [14]. DIAAS has the benefit of considering individual amino acids as nutrients, rather than a single protein value in PDCAAS, and focuses on using individual amino acid digestibilities determined at the terminal ileum, rather than fecal nitrogen digestibility. While, in certain cases, crude protein digestibility is similar to the digestibility of individual amino acids, this is frequently an overestimation when plant proteins are considered [23]. Similar to the reliance of PDCAAS on correcting for endogenous protein present in the feces, DIAAS also includes a correction factor for endogenous amino acids present in the ileal digesta. Determining this endogenous value has previously required feeding with a protein-free diet, which results in a physiologically altered state for the animal [24], as mentioned regarding PDCAAS; however, there have been more recent suggestions including combining the use of enzymatically hydrolyzed casein and ultrafiltration for the analysis of endogenous protein at the terminal ileum [25,26]. It is also worth noting that the collection of digesta from the terminal ileum is commonly an invasive procedure when compared to the fecal collection required by PDCAAS. As there is the necessity of determining the amino acid composition of the ileal digesta, there is also an increased sample analysis required for DIAAS calculation compared to PDCAAS. 

**Table 2 foods-12-02816-t002:** Benefits and detriments of PER, PDCAAS, and DIAAS as measurements of protein quality.

Protein Quality Measurement	Benefits	Detriments
PER	Only requires diet intake and weight gain to calculate [16] Measures growth rate per unit protein consumed [16]	Provides information relative to casein diets [16] Assumes all diet intake is directed towards growth [16] Rodent-based assay [16]
PDCAAS	Detailed information on amino acid composition and protein digestibility [7] Mathematically accessible for theoretical blend development [7]	Requires truncation to 1.00 (100%) after calculation for ingredients and final mixed meals [22] Use of fecal protein digestibility [7] Protein-free diets used to control for endogenous protein loss [7] Rodent-based assay [7]
DIAAS	Detailed information on amino acid composition [14] Considers amino acids as individual nutrients [14] Determination of digestibility at the terminal ileum [14] Suggested humans as the test organisms [14]	Invasive procedures required for digesta collection at the terminal ileum [14] Increased number of analytical samples compared to PDCAAS [14] Use of a protein free diet for endogenous protein loss [24] Requires truncation to 1.00 (100%) after calculation for mixed meals [14] Use of humans increases cost and tends to reduce sample size [14]

### 2.5. Alternative Options for the Regulation of Protein Content Claims

PER, PDCAAS, and DIAAS require the use of animal experimentation to determine protein quality, and therefore any regulations regarding protein content claims requiring these measurements will also mandate animal experiments. There are, however, jurisdictions that have regulations in place for determining protein content claims that do not require animal experiments, and instead rely on laboratory methods for regulatory claims. In the European Union, there is a two-tiered system for protein nutrition claims [27]. This system focuses on the energy value (i.e., caloric contribution) of a food and allows for a ‘source of protein’ claim if the energy value provided by protein is at least 12% of the total, and a ‘high protein’ claim if the energy provided by protein is at least 20% of the total. In Australia and New Zealand, the requirements for content claims are based on the total protein content of a food [28]. Under those regulations, a protein claim can be made if a food contains between 5–9.9 g of protein per serving, and if the protein content is at least 10 g per serving the food can be considered a ‘good source’ of protein. These two examples of jurisdictions that regulate protein claims using only chemical analyses highlight the capability of providing appropriate nutritional information to consumers in a manner that does not rely on animal experimentation. Such systems, however, do not consider the quality of a given source of protein.

### 2.6. Amino Acid Composition and Protein Quality of Protein-Containing Fractions

There is a wide range of methods for the generation of high protein-containing fractions of plant flours; the accurate determination of the overall amino acid composition of the resulting product is essential. In most cases, the protein of interest is hydrolyzed into its constituent amino acids by incubating a sample in hydrochloric acid at 110 °C overnight, which disrupts the peptide bonds, thereby liberating the amino acids for derivatization. This particular method is useful for all amino acids except for the sulfur amino acids, which are sensitive to acid hydrolysis and must be converted to alternative forms (cysteic acid and methionine sulphone) before hydrolysis, and tryptophan, which is liberated from the polypeptide using alkaline hydrolysis as the indole ring is sensitive to strong acids. While these methods for protein hydrolysis are relatively standardized, direct comparisons between results in the scientific literature can be complicated by the alternatives available for derivatizing amino acids. For example, the common methods used to derivatize amino acids include o-pthalaldehyde (OPA), ninhydrin, phenylisothyocyanate (PITC), and commercially available kits including AccQ-Tag. Compounding this potential variation are the multiple ways in which amino acid data can be presented, including the percent of total amino acids, g/100 g as-is basis; g/100 g protein; and g/kg dry weight, among others.

A factor that should be considered when examining amino acid composition data is the underlying conditions under which the protein-enriched fraction was produced, especially with respect to sulfur amino acids. As mentioned, the standard method for the determination of cysteine and methionine is to completely oxidize the protein and measure the total contents of cysteic acid and methionine sulfone, the resulting oxidized forms which are unusable for biological work [29]. The underlying assumption in this assay is that there was no methionine sulfone or cysteic acid present in the original sample, which has been shown to be incorrect [30,31]. As that is the case, it is probable that the standardized methodology for amino acid analysis may be overestimating the nutritionally available sulfur amino acid content in processed foodstuffs. Investigating the content of cysteic acid and methionine sulfone before and after the oxidation of a protein would provide useful information regarding the true amount of nutritionally viable sulfur amino acids. 

## 3. Effects of Pulses on Human Health

An unhealthy diet is a significant risk factor for chronic diseases, which account for more than two-thirds of all deaths worldwide [32,33,34,35]. Risk factors (such as hyperglycemia, hyperlipidemia, and overweight/obesity) can be lessened by consuming a healthy diet [36]. According to the available data, increasing the public’s intake of pulses could be a viable and affordable strategy for halting the obesity epidemic and preventing several chronic illnesses, including type 2 diabetes, cardiovascular disease (CVD), and several cancers [37]. However, the intake of pulses is surprisingly low, particularly in developed countries, despite the numerous health benefits [38] and significant savings on annual health care [38] that the regular consumption of pulses could give [39,40]. Certain attributes of pulse consumption that are beneficial to human health are listed in Table 3.

### 3.1. Cardiovascular Disease (CVD)

Nearly half of all deaths from non-communicable diseases are currently caused by cardiovascular disease (CVD), which is also the world’s most common cause of mortality [41]. Obesity, sedentary behaviour, smoking, poor diet, and alcoholism are CVD’s critical behavioural risk factors [42,43,44]. Lifestyle changes are considered the most significant preventative measures among the several methods employed to decrease the adverse consequences of CVD [42,43,44]. Healthy-eating strategies have gained the most scientific attention primarily because of the beneficial effects of plant-based diets on CVD health [45] and lowering overall mortality [46].

The available evidence suggests that eating pulses may lower CVD biomarkers and therefore reflect a dietary preventative measure [47]. According to one study [48] that specifically looked into the connection between bean consumption and the risk of CVD, eating one serving of beans per day—which is equal to 1/3 cup of cooked beans—was linked to a 38% lower risk of myocardial infarction (MI) (odds ratio (OR) = 0.62, 95% confidence interval (CI) 0.45–0.88; *p* 0.05) compared to people who ate beans infrequently. However, eating more than one serving of beans per day did not offer any further protection against MI occurrence. Other studies link a general legume diet to an increased risk of CVD. During a 19-year cohort study by Bazzano et al., 3680 incident instances of CVD and 1802 coronary artery disease (CAD) cases were recorded among more than 9000 patients without CVD [49]. In their follow-up study, they found that when compared to people who consumed legumes less than once per week, those who consumed legumes four or more times per week had a 22% lower risk of coronary heart disease (CHD) (RR = 0.78, 95% CI 0.68–0.90) and an 11% lower risk of CVD (RR = 0.89, 95% CI 0.80–0.98) [49].

Furthermore, according to data from the National Health and Nutrition Examination Survey (NHANES), legumes (peanuts, dry beans, and peas) had a robust and independent inverse relationship with the risk of CHD and CVD [49]. The consumption of beans of darker colours, such as black and red, showed a more significant decrease in parameters related to vascular tone when compared to the consumption of beans of lighter colours, such as brown and white, after 2 h of ingestion, according to an evaluation of cardiovascular changes such as pulse wave velocity, blood pressure, and low-density lipoprotein (LDL) cholesterol, stimulated by the consumption of various types of beans (1/4 cup) and rice (3/4 cup) [50]. 

Additionally, consuming pulses positively impacts CVD-contributing factors such as BMI and C reactive protein levels (CRP) [36]. Bean eaters, for instance, were shown to have lower body weights (*p* = 0.008) and smaller waist measurements (*p* = 0.043) in comparison to non-consumers in the National Health and Examination Survey (NHANES; 1999–2002), as well as a trend towards lower systolic blood pressure [51]. Moreover, extruded dry beans (91.9 g/day) were found to significantly lower plasma plasminogen activator inhibitor-1 (PAI-1) levels in a randomized, cross-over investigation, while cholesterol levels were unaffected [52]. Overall, the data point to increased pulse consumption as a dietary measure to help reduce CVD risk [53]. 

### 3.2. Satiety

Satiety can be considered the feeling of ‘fullness’ after consuming a meal, with the duration of that feeling varying widely between individuals and being dependent on meal size and composition. Studying the effects of protein sources on satiety aligns with attempts to reduce the overall caloric intake, facilitating weight maintenance or loss, a target outcome in societies where obesity has a high prevalence rate. Increasing the percentage of calories consumed as protein can lead to weight loss [54], while increasing protein intake may reduce caloric intake via increased satiety [55,56,57].

While these studies have focused on animal-based proteins in the diet, a study comparing the effect of soy and beef consumption on appetite, satiety or food intake determined that there was no difference between these two protein sources [58]. Pulses increased satiety in a study comparing rice, wheat, and rice/pulse meals [59], while the consumption of lentils reduced food intake compared to a pasta meal [60]. A randomized crossover trial investigating the effect of pea protein and a combination of pea protein and hull fiber on appetite and food intake found no differences, suggesting that increases in satiety and the suppression of appetite are pulse-specific effects [61]. In addition to protein source, the timing of certain dietary interventions can also differentially affect satiety, as the ingestion of 20 g of casein or pea protein 30 min before a meal reduced food intake compared to whey protein [62]. 

### 3.3. Lipid Metabolism

The dysregulation of lipid metabolism, as can be indicated by hyperlipidemia, is one of the major factors that can induce atherosclerotic cardiovascular disease [63]. A meta-analysis was recently conducted on randomized control trials investigating the effect of substituting dietary animal-based protein for plant protein [64]. This study concluded that it was beneficial to replace animal-based protein with plant protein in general; however, it was cautioned that specific outcomes such as low-density lipoprotein C, non-high-density lipoprotein C, and apolipoprotein B were variable between studies included in the analysis. Perhaps the most well-studied plant protein source, with respect to lipid metabolism, is soy [65,66,67,68]. The evidence surrounding soy consumption and lowered cholesterol levels has been prevalent enough that in 1999 the FDA approved a health claim regarding soy protein consumption and a reduced risk of coronary disease [67]. However, conclusions from an increasing number of observational studies focused on the relationship between soy consumption and the risk of CHD have been inconsistent [69,70]. There has been some discussion regarding the mechanisms by which soy exerts its hypocholesterolemic effect, primarily regarding whether the protein or isoflavone component is more important [71,72]. It has since been demonstrated that it is the protein fraction that is capable of lowering serum lipid levels [73]. Interestingly, the intact protein is required for cholesterol reduction, and not simply a mixture of similar amino acids [74]. This is supported by the determination that an isolated 7S globulin α’ subunit of soy protein was able to upregulate liver lipid receptors, and reduce plasma cholesterol and triglycerides [75]. Other plant proteins such as lupin [76,77,78], faba bean [79], pea [80,81], and chickpea [82] have also been investigated for their effect on cholesterol metabolism. The consumption of the plant protein was beneficial in all cases, leading to reduced cholesterol levels. It has also been determined that heating applied during processing or cooking does not reduce the hypocholesterolemic properties of certain plant proteins such as pea [81] and chickpea [82].

Purified chickpea and lentil proteins, 92% and 90% protein, respectively, were able to decrease plasma very-low-density lipoprotein and reduce plasma triglyceride when compared to a casein control [83], thereby further indicating that processing does not necessarily reduce the efficacy of plant proteins on lipid metabolism. Furthermore, several studies show that eating more pulses can lower high blood cholesterol levels. In a meta-analysis of ten studies with high non-soy legume intake, total cholesterol (−11.76 mg/dL), LDL cholesterol (−7.98 mg/dL), triglycerides (−18.94 mg/dL), and high-density lipoprotein (HDL) cholesterol (0.85 mg/dL) all showed benefits [84]. In a 2-month hypocaloric-diet-based study of obese adults eating four servings of legumes per week, Hermsdorf et al. found significant decreases in total cholesterol concentrations (215 ± 27 mg/dL vs. 182 ± 27 mg/dL, *p* < 0.05) and systolic blood pressure (115 ± 13 mmHg vs. 106 ± 10, *p* < 0.05) [85]. Another study demonstrated that after nine weeks, atherogenic mice treated with a fresh-ground bean protein hydrolysate—roughly the same amount of cooked beans consumed daily—showed a 6% reduction in serum total cholesterol and a 9% reduction in serum LDL-cholesterol (*p* < 0.05) in plasma triglycerides and total cholesterol. The bioactive components included in pulse protein ingredients, such as lectins, hemagglutinins, and enzyme inhibitors, may have health benefits like lowering serum glucose levels and reducing obesity [86,87,88]. Histidine, tryptophan, and phenylalanine were found in peptides from chickpea legumins, and these amino acids have an antioxidant potential [89]. According to Li et al. [90], the fraction of chickpea protein hydrolysate (fra. IV) with the highest levels of hydrophobicity (125.62 kcal/mol amino acid residue) and total hydrophobic amino acids (38.94%) also had the best antioxidant activity. The inhibition of the angiotensin I converting enzyme, antioxidant capacity, antibacterial, and anticancer properties was present in the hydrolysates and peptides of common bean and mung bean proteins [91,92]. Additionally, compared to the atherogenic diet alone, significant reductions in inflammation and endothelial dysfunction were shown by 62% higher endothelial nitric oxide synthase (e-NOS) levels and 57% higher nitric oxide serum concentration, as well as gene expression changes in tumor necrosis factor (TNF, 94% reduction) and angiotensin II (79% reduction) [93]. Due to these nutraceutical qualities, adding pulse proteins to diets may be advised for managing cholesterol and triglycerides and lowering the risk of chronic illnesses like hypertension, cardiovascular disease, and cancer.

### 3.4. Diabetes Mellitus

Diabetes is becoming more prevalent in adults and children worldwide [33]. More than 90% of instances of diabetes are type 2 (T2D), and having a high BMI (<25 kg/m^2^) is one of the main risk factors. Therefore, adopting healthy lifestyle habits that promote a healthy body weight, and, among them, consuming a well-balanced diet can help to prevent T2D and reduce its effects [36]. Thus, frequent pulse consumption can potentially aid in controlling and preventing diabetes.

Numerous health organizations recommend consuming pulses or legumes, in general, to assist individuals with insulin resistance and/or diabetes in regulating their blood glucose levels. Epidemiological studies show a negative correlation between bean consumption and the frequency of chronic disorders such as T2D [94]. However, legumes and beans have been proven to reduce systolic blood pressure (−4.5 mmHg, *p* < 0.001) and glycated hemoglobin (HbA1c, −0.5%) outcomes in T2D when ingested alone, as a component of low glycemic index (GI) diets or diets richer in fibre [95]. The amount of glycosylated proteins (such as fructosamine and HbA1C) in the blood increases with chronic blood glucose elevation [36]. Blood concentrations of these glycated proteins are a longer-term indicator of blood glucose control. In a 2009 meta-analysis, Sievenpiper et al. [96] examined 41 randomized controlled trials that included pulses either alone or as a part of a low-GI or high-fibre diet to determine the relationship between the consumption of pulses and several markers of glycemic control. The researchers reported evidence that pulses alone or as part of a low-GI or high-fibre dietary intervention improved several markers of glycemic control (i.e., lowered fasting blood glucose, insulin, and glycosylated proteins). However, there was significant heterogeneity among studies [96]. Longer follow-up periods (>4 weeks) and diets that were metabolically regulated showed more significant benefits in people with diabetes. Researchers discovered that T2D patients who were randomly assigned to a low-GI legume diet experienced a more significant decrease in HbA1C (−0.5%, 95% CI −0.6% to −0.4%) than those who were assigned to a high-wheat-fibre diet [97]. According to Jenkins et al. [97], frequent pulse inclusion (1/2 cup/day; 100 g) in the diets of people with diabetes would likely result in a 0.5% reduction in HbA1C. These data suggest that the regular inclusion of pulses in the diet in place of other carbohydrate sources, as well as their high fibre content and low GI, would improve glycemic control in people with insulin resistance and diabetes.

### 3.5. Muscle Protein Synthesis

The effect of protein consumption on muscle growth is of interest to a wide variety of groups, ranging from those focused on sports nutrition to those involved in developing dietary plans to prevent or combat sarcopenia. It is well known that resistance exercise can induce muscular hypertrophy [98]. However, recent research has been directed towards the effect of particular amino acids on muscle protein synthesis, leucine in particular [99,100,101]. A recent study investigated the effect of leucine on myofibrillar protein synthesis in which it was determined that increased protein leucine consumption increased myofibrillar protein synthesis when compared to isonitrogenous, isocaloric controls [102]. This has led to the concept of a leucine threshold, where once intracellular leucine reaches a certain concentration, muscle protein synthesis is stimulated [103]. Considering that most plant proteins have a lower leucine content than animal proteins [103,104], it is valid to question whether plant-based protein consumption is comparable to muscle protein synthesis. In addition to differences in leucine content, the fact that plant proteins are generally limiting in other essential amino acids and have lower protein digestibility are also factors that impact aminoacidemia, and thereby the potential for plant proteins to impact muscle protein synthesis. The methods proposed to overcome these limitations include increasing the quantity of material consumed or manipulating the amino acid composition via fortification with specific amino acids or through blending different plant protein sources [104]. A study performed in rodents determined that the consumption of greater quantities of plant protein, wheat in this case, induced similar protein synthesis rates as whey [105]. However, a comparison between protein synthesis in elderly males after soy and whey protein isolate ingestion still determined that soy isolate was relatively ineffective in stimulating muscle protein synthesis compared to whey [106]. A similar result was found in another study investigating sarcopenia, where it was seen that the consumption of animal-based proteins may have a greater efficacy in preventing sarcopenia than a similar quantity of soy protein [107]. This brief discussion of the role of protein consumption in the growth and maintenance of muscle mass contains only a few examples from the literature, although it highlights the fact there are many instances where distinctions in the amino acid composition of protein sources have led to differences in muscle protein synthesis.

### 3.6. Gut Health

There are several implications for public health in the rapidly developing field of study of how diets affect the gut microbiome. Numerous diseases, such as diabetes, cardiovascular disease [108], inflammatory bowel disease (IBD), and colorectal cancer (CRC) [109], have been associated with dysbiosis of the gut microbiota. In contrast, evidence indicates that eating whole foods, rather than refined, processed meals, is linked to a healthy gut microbiome and a low prevalence of diet-related disease [110], emphasizing the possibility that eating pulses regularly may enhance health. 

Preclinical studies have looked at the effects of feeding mice diets enriched with different pulses, such as cranberry bean [111], chickpea [112], white kidney bean [113], pinto bean [114], navy bean [115], and black bean [116]. Prior research has provided evidence that consuming pulses changes the microbiota, for example, increasing the abundance of the health-promoting *Akkermansia muciniphila*, and improving several gut health indicators, including the expression of genes associated with improved gut barrier function and short-chain fatty acid (SCFA) production. Black and navy beans increased the biomarkers of colon barrier integrity including microbial carbohydrate fermentation and decreased protein fermentation, as shown by higher SCFAs and decreased branched-chain fatty acids (BCFAs) in mice, including improved mucus epithelial barrier integrity and lower permeability [117]. IBD, colitis, and colon cancer are gastrointestinal disorders characterized by chronically dysregulated inflammatory response pathways, which SCFAs have a high potential to reduce [118]. Zhang et al. [118] investigated the effects of bean flour (whole cooked, freeze-dried) in a mouse colitis model, showing that black bean and navy bean supplementation reduced colitis-related indicators of inflammation through bioactive components, namely fermentation-derived SCFA and/or phenolic compounds. 

However, not all pulses exhibited the same effects; for example, mice fed lentil and bean diets saw increases in the abundance of *Akkermansia muciniphila* in their cecal abundance, but mice fed chickpea and dry-pea diets did not [119]. A different research team discovered that animals fed a black-bean diet had more significant improvements in indicators of intestinal barrier integrity than mice fed navy beans [116]. Diverse pulse types and processing techniques, as well as variations in prebiotic carbohydrate content between market classes, may affect the gut microbiota differently [120,121]. Other substances that are difficult to digest, such Bowman–Birk family protease inhibitors, may also have advantageous effects on gastrointestinal health [122].

**Table 3 foods-12-02816-t003:** Health benefits associated with pulse consumption.

Health Benefit	Description
Cardiovascular disease (CVD)	Lower CVD biomarkers [47,50]. Reduce the risk of myocardial infarction [48]. Lower the risk of coronary heart disease [49]. Decrease the risk of cardiovascular disease [47,48,49,50].
Satiety	Provide a feeling of fullness [54]. Reduce overall caloric intake [55,56,57].
Lipid metabolism	Lower cholesterol levels [73,74,75,76,77,78,79,80,81,82,83,84,85]. Reduce the risk of atherosclerotic cardiovascular disease [67,69,70].
Diabetes mellitus	Regulate blood glucose levels [96]. Improve glycemic control [97]. Reduce the risk of type 2 diabetes [36].
Muscle mass	Aid muscle protein synthesis [104,105,106]. Support muscle growth [107].
Gut health	Enhance gut microbiota [120,121,122]. Improve gut barrier function [116,117]. Increase production of short-chain fatty acids (SCFAs) [118]. Potentially reduce inflammation [118].

## 4. Conclusions

There is a trend towards the increased consumption of alternative protein sources, such as plant-based protein, in the developed world. Protein sources such as beans, peas, lentils, chickpeas, and cereals are lower in protein content than animal sources, However, the environmental impact of plant protein production is much lower than that of livestock. Protein sources, including plant proteins, are not equivalent in total protein content or in overall quality. In order to understand the quality of these sources, an accurate assessment of their inherent characteristics, such as their ability to induce growth, their specific amino acid profile, and overall digestibility, is vital. The current methods being used, PER/PDCAAS/DIAAS, have their advantages and disadvantages. However, adherence to the required method for the jurisdiction of interest, PER for Canada and PDCAAS for the United States, is necessary for a successful protein content claim. The protein contents of pulse and cereal crops position these as viable sources of high-quality protein, while the generation of protein concentrates and protein isolates enhances the capacity for inclusion within novel consumer products. In addition to the nutritive qualities of plant-based proteins, there are also beneficial properties for human health when compared to animal protein sources such as a reduction in cholesterol, a lower incidence of cardiovascular disease, increased satiety, and the potential for novel therapeutics in the form of drug delivery mechanisms using plant proteins. While, frequently, the quality of a plant protein is reduced to the score generated by the appropriate analytical method, with further understanding it becomes apparent that with potential effects on the environment, the overall nutritive value, and the impact on human health outcomes, protein quality should be considered as more than just a single number. 

## Figures and Tables

**Figure 1 foods-12-02816-f001:**
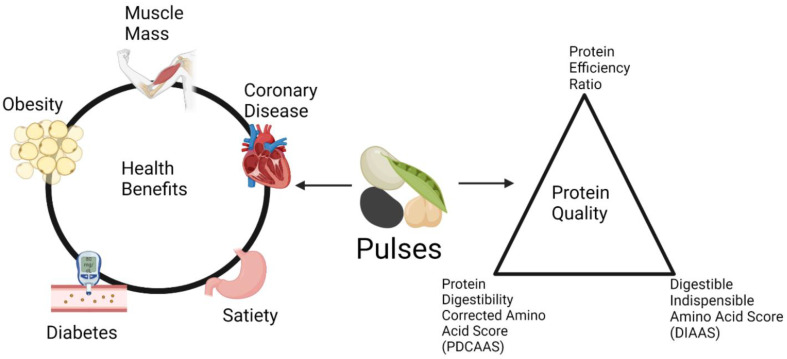
Health benefits of pulse consumption and different measures of protein quality. Created with bioRender.

**Table 1 foods-12-02816-t001:** Adjusted PER, PDCAAS, and DIAAS values for selected animal and plant protein sources.

	Adjusted PER	PDCAAS	DIAAS
Milk ^a^	2.50	1	114
Eggs ^a^	3.10	1	113
Chicken ^a^	2.70	1	108
Oatmeal ^a^	1.80	0.82	84
White bread ^a^	1.00	0.28	29
White Rice ^a^	1.50	0.56	57
Tofu ^a^	2.30	0.56	52
Red Kidney Beans ^b^	1.55	0.55	51
Navy Beans ^b^	1.51	0.67	65
Whole Green Lentils ^b^	1.30	0.63	58
Split Red Lentils ^b^	0.98	0.54	50
Split Yellow Peas ^b^	1.42	0.64	73
Split Green Peas ^b^	0.86	0.50	46
Black Beans ^b^	1.61	0.53	49
Chickpeas ^b^	2.32	0.52	85
Pinto Beans ^b^	1.64	0.59	60

^a^ Modified from [18]. ^b^ Modified from [19].

## Data Availability

No new data were created or analyzed in this study. Data sharing is not applicable to this article.

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
