# Peer review of "Plant Proteins: Methods of Quality Assessment and the Human Health Benefits of Pulses"

_foods, 2023, doi:10.3390/foods12152816_

Round 1
Reviewer 1 Report
The manuscript is well written. The topic is interesting. There are only some details to check.
-Page 5 Line 12: Use italics "in vivo".
-Page 7 Line 6: Use "CVD" instead of "cardiovascular disease".
-Page 7 Line 7: Check punctuation " Bazzano, et al.,,".
-Page 7 Line 11: Explain the full name of "CHD" when first used.
-Page 8 Line 13: Explain the full name of "CD" when first used.
-The number of references in the text is 121, but 122 appear in the "References", please check.
-Check the format of references.
-Check the format of the references.
Author Response
-Page 5 Line 12: Use italics "in vivo".
-Page 7 Line 6: Use "CVD" instead of "cardiovascular disease".
-Page 7 Line 7: Check punctuation " Bazzano, et al.,,".
-Page 7 Line 11: Explain the full name of "CHD" when first used.
-Page 8 Line 13: Explain the full name of "CD" when first used.
We have made the suggested changes to the text of the manuscript.
-The number of references in the text is 121, but 122 appear in the "References", please check.
Thank you for your comment. Reference 122 is located in the footnote of Table 1.
Reviewer 2 Report
The paper of Nosworthy et al. is interesting and is written in a nice manner. Anyway, some changes are suggested before final decision to be made:
The title of the paper should be reconsidered as it is too general.
The abbreviations must be defined only at first use in the manuscript body. Take for instance PER which was defined twice at page 2, PDCAAS etc.
I suggest inserting a graphical representation to present the advantages and disadvantages of PER, PDCAAS and DIAAS. An image would help the reader to easier get the whole picture.
The heading “3. Section 3: Effects of Pulses on Human Health” needs to be reconsidered. Remove “Section 3:”
Pay attention to punctuation on the last line of page 6.
Provide the meaning of OR and CI on the first paragraph at page 7.
A table depicting the main benefits provided by the pulses consumption on human health would highly benefit the reader. The authors should consider inserting a such table in the section 3. Effects of Pulses on Human Health
Author Response
The paper of Nosworthy et al. is interesting and is written in a nice manner. Anyway, some changes are suggested before final decision to be made:
The title of the paper should be reconsidered as it is too general.
We thank the reviewer for their comment. However as this is an overview of protein quality assessment and discussion of how plant proteins can be beneficial to human health outcomes the authors believe the current title to be appropriate for the material discussed.
The abbreviations must be defined only at first use in the manuscript body. Take for instance PER which was defined twice at page 2, PDCAAS etc.
We have removed the additional abbreviation definitions.
I suggest inserting a graphical representation to present the advantages and disadvantages of PER, PDCAAS and DIAAS. An image would help the reader to easier get the whole picture.
To accommodate this suggestion as well as those from other reviewers we have included a new table listing the advantages and disadvantages of PER, PDCAAS, and DIAAS.
The heading “3. Section 3: Effects of Pulses on Human Health” needs to be reconsidered. Remove “Section 3:”
We agree and have removed “Section 3” from the heading.
Pay attention to punctuation on the last line of page 6.
We have modified that sentence for clarity.
Provide the meaning of OR and CI on the first paragraph at page 7.
We have added those details to the appropriate paragraph.
A table depicting the main benefits provided by the pulses consumption on human health would highly benefit the reader. The authors should consider inserting a such table in the section 3. Effects of Pulses on Human Health.
We agree and have included a new table detailing the effects of pulses on human health.
Reviewer 3 Report
I would advise the authors to make this interesting paper more readable. Hence, 2 tables are required to present this knowledge and make some comparisons. One on section 2 2.4. Advantages and Disadvantages of PER, PDCAAS and DIAAS and another one on Section 3: Effects of Pulses on Human Health. The authors need to compare referring to significant parameters in these studies. Moreover, a figure would be relevant.
English is not a problem in this manuscript
Author Response
I would advise the authors to make this interesting paper more readable. Hence, 2 tables are required to present this knowledge and make some comparisons. One on section 2 2.4. Advantages and Disadvantages of PER, PDCAAS and DIAAS and another one on Section 3: Effects of Pulses on Human Health. The authors need to compare referring to significant parameters in these studies. Moreover, a figure would be relevant.
Thank you for your comments. We concur and have added additional tables to provide better clarity. This includes Table 2: Benefits and Detriments of PER, PDCAAS, and DIAAS as measurements of Protein Quality and Table 3: Health Benefits Associated with Pulse Consumption.
Round 2
Reviewer 2 Report
The manuscript was improved and can be accepted for publication.
Author Response
Thank you for providing your comments on this work.
Reviewer 3 Report
the authors have modified the manuscript, however a figure has not been added. Please provide a relevant figure.
Author Response
We have generated a figure describing the theme of this review as it relates to health benefits and protein quality of pulses.